# The environmental risk assessment of cell-processing facilities for cell therapy in a Japanese academic institution

Mitsuru Mizuno[1,2], Kentaro Endo[1], Hisako Katano[1,2], Ayako Tsuji[2], Naomi Kojima[1], Ken Watanabe[1], Norio Shimizu[1,2], Tomohiro Morio[2,3], Ichiro Sekiya[1,2]*

1 Center for Stem Cell and Regenerative Medicine, Tokyo Medical and Dental University, Bunkyo-ku, Yushima, Tokyo, Japan, 2 Center for Transfusion Medicine and Cell Therapy, Tokyo Medical and Dental University, Bunkyo-ku, Yushima, Tokyo, Japan, 3 Department of Pediatrics and Developmental Biology, Graduate School, Tokyo Medical and Dental University, Bunkyo-ku, Yushima, Tokyo, Japan

* sekiya.arm@tmd.ac.jp

## Abstract

Cell therapy is a promising treatment. One of the key aspects of cell processing products is ensuring sterility of cell-processing facilities (CPFs). The objective of this study was to assess the environmental risk factors inside and outside CPFs. We monitored the temperature, humidity, particle number, colony number of microorganisms, bacteria, fungi, and harmful insects in and around our CPF monthly over one year. The temperature in the CPF was constant but the humidity fluctuated depending on the humidity outside. The particle number correlated with the number of entries to the room. Except for winter, colonies of microorganisms and harmful insects were detected depending on the cleanliness of the room. Seven bacterial and two fungal species were identified by PCR analyses. Psocoptera and Acari each accounted for 41% of the total trapped insects. These results provide useful data for taking the appropriate steps to keep entire CPFs clean.

## Introduction

Regenerative medicine using stem cells holds great promise for the treatment of some incurable diseases. In particular, mesenchymal stem cells (MSCs) [1] and induced pluripotent stem cells (iPSCs) [2] have been considered feasible cell sources. In recent years, many researchers have conducted clinical trials for developing new cell products [3–5]. In Japan, cell products are processed at each academic cell-processing facility (CPF) based on a new regulatory framework for regenerative medicine [6, 7]. The safety of cell products is threatened not only by undesired cell characteristics, such as tumorigenicity, but also by cell culture contamination caused by microorganisms. Sterilization is the simplest way to eliminate such infections; however, living cell products cannot be sterilized. Therefore, the most important aspect of cell processing safety is guaranteeing the sterility of the process.

The sterility of cell products is secured by the appropriate management of CPF environments and traceability based on these records. Such environmental management policy should

**Data Availability Statement:** All relevant data are within the paper and its Supporting Information files.

**Funding:** The Research Project for Practical Applications of Regenerative Medicine (JP18bk0104065) supported IS from the Japan Agency for Medical Research and Development (AMED) to complete this study.

**Competing interests:** The authors have declared that no competing interests exist.

be planned to depend on CPF structure to maintain proper cleanliness and eliminate the risk of extrinsic contamination of the process [8, 9]. The risk of extrinsic contamination can be affected by the location of the CPF, the environment inside or outside the CPF, and the degree of human intervention during the process [10, 11]. These environmental risk factors include temperature, humidity, number of particles, entrance time to a clean room, airborne microorganisms, and harmful insects. In Japan, the temperature and humidity fluctuate according to the season. Therefore, to perform an appropriate environmental risk assessment, it is essential to obtain total environmental data throughout the year. Since environmental factors may interact each other, the spatiotemporal relationships among them should be also evaluated.

The objective of this study was to assess the environmental risk factors inside and outside a CPF to guarantee the safety of cell products. We collected comprehensive environmental data throughout the year during an ongoing clinical study and investigated the relationships among them. Their effects on the sterility of processed cell products were also assessed.

## Material & methods

### Classification of areas in the cell-processing facility (CPF)

The CPF was divided into uncontrolled and controlled areas (Fig 1A). The uncontrolled area was defined as the general environment (GE) and the controlled areas were divided into four areas (Grade D, Grade C, Grade B, and Grade A) according to cleanliness defined by "consideration on aseptic operation in cell culture processing facilities" based on the "Safety Act" published by the Japanese Society for Regenerative Medicine (S1 Table). Grade A was an area where aseptic operation was performed.

Fig 1B shows the schematic diagram of our CPF. Operators entered from the entrance, went through the first gowning room, the buffer room, and the second gowning room in that order, entered the cell-processing room, and worked using a safety cabinet. The operators then passed through the second de-gowning room, the buffer room, and the first de-gowning room before exiting.

In this study, in front of the facility, the corridor and workspace outside the CPF, and the laboratory outside the CPF were considered the GE. The entrance of the CPF was labeled Grade D; the first gowning room, the buffer room, and the supply room were labeled Grade C; the second gowning room and cell-processing room were labeled Grade B; and the space in the safety cabinet was labeled Grade A.

### Data collection

Assessment items were collected from January 2018 to December 2018. The data in the CPF during a certain period in February and March were not collected due to calibration and validation of the CPF for an annual inspection. The total number of entrance times in a month was counted based on the cell-processing record.

### Monitoring of temperature and humidity

The temperature and humidity in each area were measured using the precision hair hygro-thermometer (Higest II: Sato Keiryoki Mfg. Co., Ltd., Tokyo, Japan). The temperature outside the building was based on the database of the Japan Meteorological Agency. The mean values of temperature and humidity in the Grade D, Grade C, and Grade B areas were calculated from data automatically measured every 10 minutes (TH-EV6A, Shinyei Technology Co., Ltd., Kobe, Japan).

### (a) Classification of areas in the CPF according to cleanliness

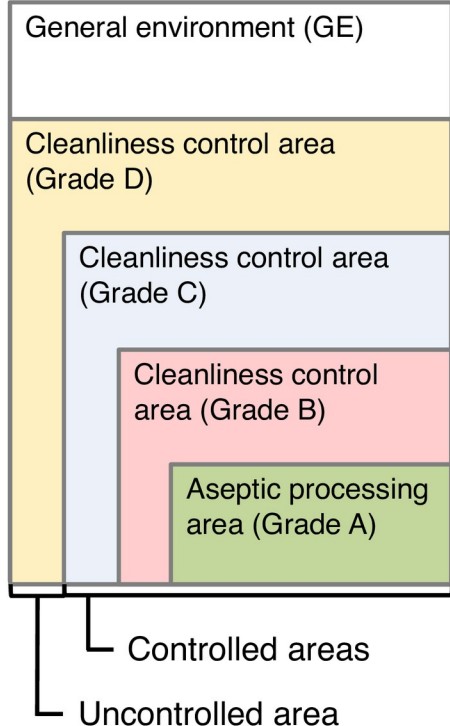

### (b) Schematic diagram of the CPF and assessment items

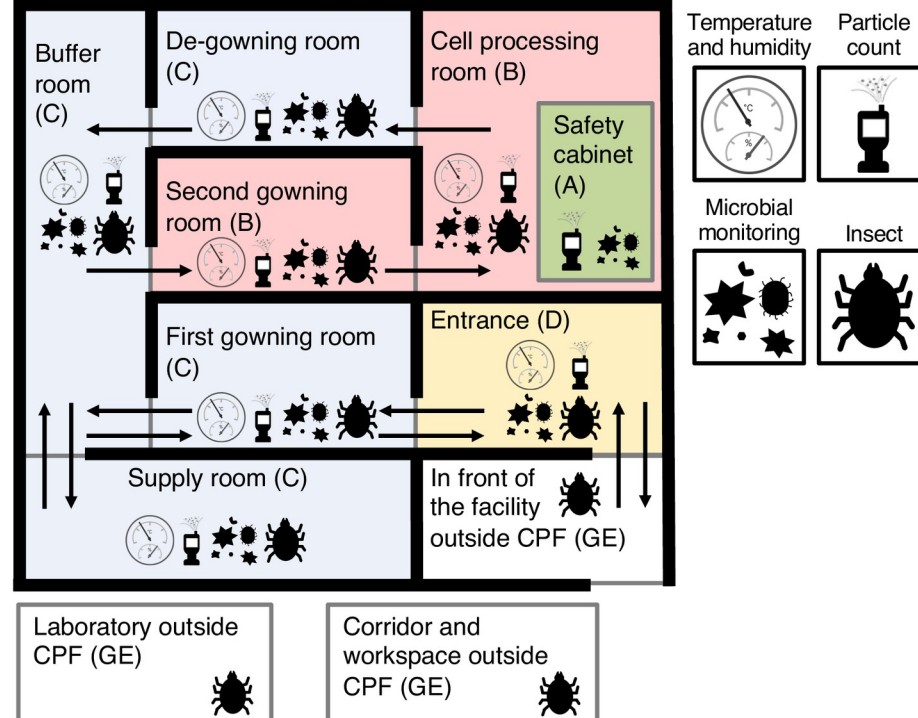

**Fig 1. Assessment items for each room of the Cell-Processing Facility (CPF).** (a) Classification of areas in the CPF according to cleanliness defined by the Japanese Society for Regenerative Medicine (S1 Table). Assessed area is divided into uncontrolled and controlled areas. The uncontrolled area is defined as the general environment (GE), and the controlled areas are divided into four areas according to cleanliness. (b) Schematic diagram of CPF and assessment items in each room. Operators enter from the entrance, go through the first gowning room, the buffer room, and the second gowning room in that order, enter the cell-processing room, and work using a safety cabinet. The operators then pass through the de-gowning room, the buffer room, and the first gowning room before exiting. The colors represent the cleanliness grades. The illustrations show assessment items for #1, "temperature and humidity"; #2, "particle count"; #3, "microbial monitoring"; and #4, "insect" performed in each room. Insects were also monitored in the corridor and workspace, laboratory, and in front of the CPF.

## Monitoring of particles

The number of particles ($\geq 0.5$ μm) per $m^3$ was manually obtained using METONE HHPC 3+ (Beckman Coulter, Inc., CA, USA) every two weeks. The maximum value was used. The integrated number of particles ($\geq 5$μm) was also obtained only in Grade B every 10 minutes using an automatic device (KA-02, Rion Co., Ltd., Tokyo, Japan). The data in cleanliness-controlled areas were shown as the maximum value in each room. Obtained data were divided in operation (during processing) and at rest (not during operation). Pearson's correlation coefficient between the number of entrance times to Grade B and the integrated number of particles in Grade B at all times including in operation and at rest was further analyzed.

## Monitoring of microorganisms

One thousand L of air was aspirated by Biosamp (Midori Anzen Co., Ltd., Tokyo, Japan), and airborne microorganisms were cultured on soybean-casein digest agar (Becton, Dickinson and Company, NJ, USA). After five days of culture, macroscopic images of colonies were captured by M165 FC (Leica Microsystems GmbH, Wetzlar, Germany). In addition, settle plate and glove print tests were performed using a soybean-casein digest agar to assess the presence of microorganisms in Grade A. The quantification of colonies was visually evaluated after five

days. The data in cleanliness-controlled areas were shown as the maximum value in each room. The chi-square test was performed to assess the association between humidity and the number of airborne microorganisms in Grade C and Grade D (test number = 133). An odds ratio was calculated for the rate of the occurrence of airborne microorganisms.

## Identification of bacterial and fungus species

Colonies of microorganisms were manually picked up and their nucleic acid extracted using a DNA extracting kit (QIAamp DNA mini kit, Qiagen, Venlo, Netherlands). Nucleic acids were amplified by PCR using a bacterial ribosomal RNA-specific designed primer and probe set [12]. Real-time PCR was performed in a LightCycler 480 instrument (Roche Diagnostics, Basel, Switzerland). For PCR reaction, the following reagents were used: forward primer; agg-cagcagtDRggaat reverse primer; ggactacYVgggtatctaat probe FAM-tgccagcagccgcggtaatacR-Dag-BHQ and the LightCycler 480 Probes Master (Roche Diagnostics). DRYV in the sequence of primer and probe represents mixed bases defined by the International Union of Pure and Applied Chemistry. The PCR products were directly sequenced with Applied Biosystems 3130xl Genetic Analyzer (Thermo Fisher Scientific, Waltham, MA, USA). Bacterial and fungus species were identified by querying NCBI Nucleotide BLAST sequence data [13]. The species suggested by BLAST with an identified rate over 99% was considered a candidate species.

## Monitoring of harmful insects

Traps (Earth Environmental Service Co., Ltd., Osaka, Japan) were settled in each area. After $31 \pm 7$ days, the number of insects per trap was counted, and the species were identified macroscopically or using a stereo microscope. The number of insects per trap in the corridor outside the CPF was compared below 15˚C and 15˚C or more outside the building by the Mann Whitney U-test.

## Cell processing and sterility test

For clinical research "intraarticular injections of synovial stem cells for osteoarthritis of the knee" (UMIN000026732), cell products were prepared in the CPF. Detailed processing protocols have been reported previously [4]. Briefly, synovium was treated with enzyme and cultured in a medium containing autologous serum and antibiotics for 14 days. Twenty million cells were administered intraarticularly. According to the regulations of Japanese Pharmacopoeia, part of the cell products was subjected to a sterility test using fluid thioglycolate medium (Becton, Dickinson and Company) and soybean-casein digest broth (Becton, Dickinson and Company). These media were incubated at $22.5 \pm 2.5$˚C and $32.5 \pm 2.5$˚C, respectively. After 14 days, the presence or absence of microorganisms was visually evaluated.

## Ethics approval and consent to participate

This clinical study (RM2018-006) was approved by the certified special committee for regenerative medicine in Tokyo Medical and Dental University (committee reference number: NA8140003) and submits a plan to the Minister of Health, Labor, and Welfare (MHLW, PB3160032). This clinical study was carried out in accordance with the Helsinki Declaration.

## Statistical analysis

All statistical analysis was performed with Prism 8 software (GraphPad Inc., La Jolla, CA, USA). Each statistical analysis is described in the material and methods section and figure legends. Two-tailed p values smaller than 0.05 were considered statistically significant.

## Results

### Monitoring of temperature, humidity, and number of particles

Although the temperature outside the building changed with seasons, that in the CPF was constant throughout the year (Fig 2). The temperature in Grade B and Grade C was always a few degrees lower than that in Grade D. The humidity in Grade B–D varied largely depending on that outside the building with highs in summer and lows in winter.

### Monitoring of particles

Monthly transitions in the number of particles ($\geq$ 0.5μm) differed from Grade B–D (Fig 3A). In Grade A, the particle number was almost zero except for February, when it was not measured. The number of particles was well below the acceptable limit, which was under 3,520 for Grade A and B, 35,200 for Grade C, and 3.52 million for Grade D. The integrated number of particles ($\geq$ 0.5μm) in Grade B correlated with the integrated number of entries to the cell-processing room (Fig 3B, R = 0.68, p = 0.03). The integrated number of particles ($\geq$ 5μm) in Grade B was also correlated with the integrated number of entries to the cell-processing room (S1 Fig, R = 0.64, p = 0.048).

### Monitoring of microorganisms and identification of species

Monthly transitions of the colony number of airborne microorganisms differed from Grade B–D (Fig 4A). In Grade A, the colony number was completely zero except for February, when

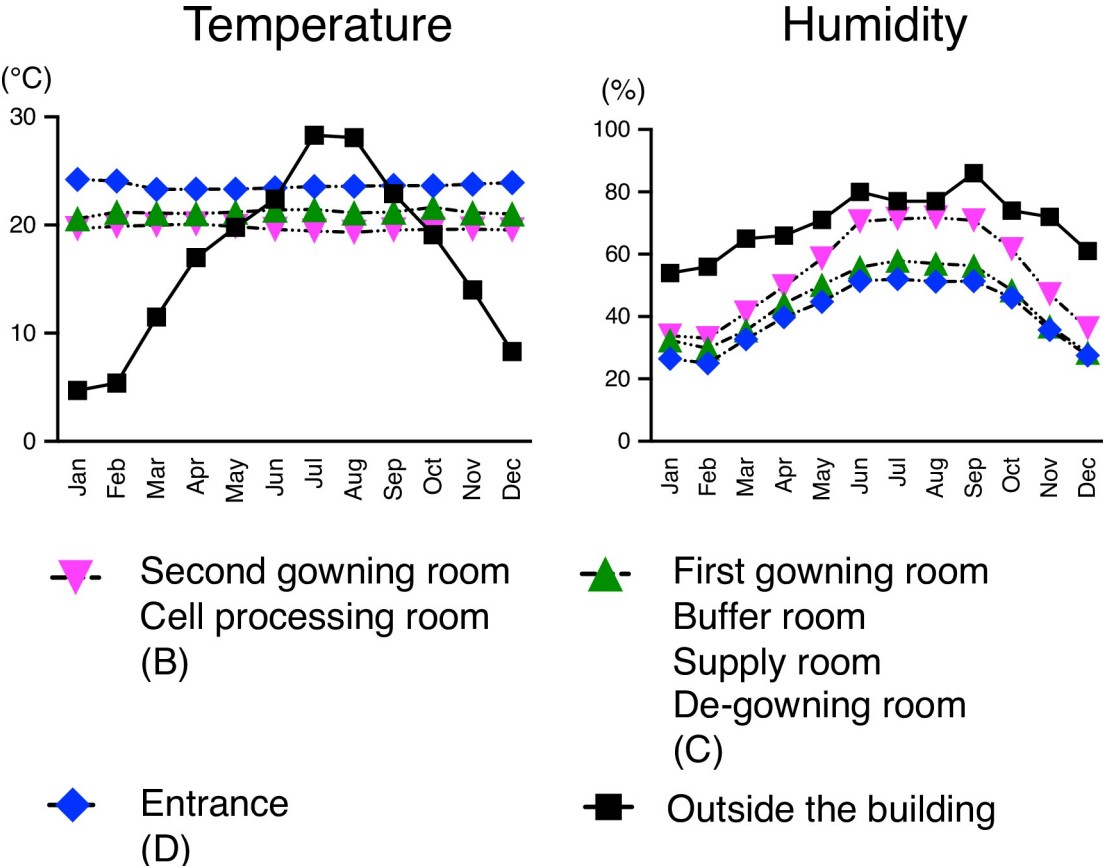

**Fig 2. Monthly transitions of temperature and humidity.** The temperature outside the building is based on the database of the Japan Meteorological Agency. The data in cleanliness-controlled areas are shown as the average values of each measurement point.

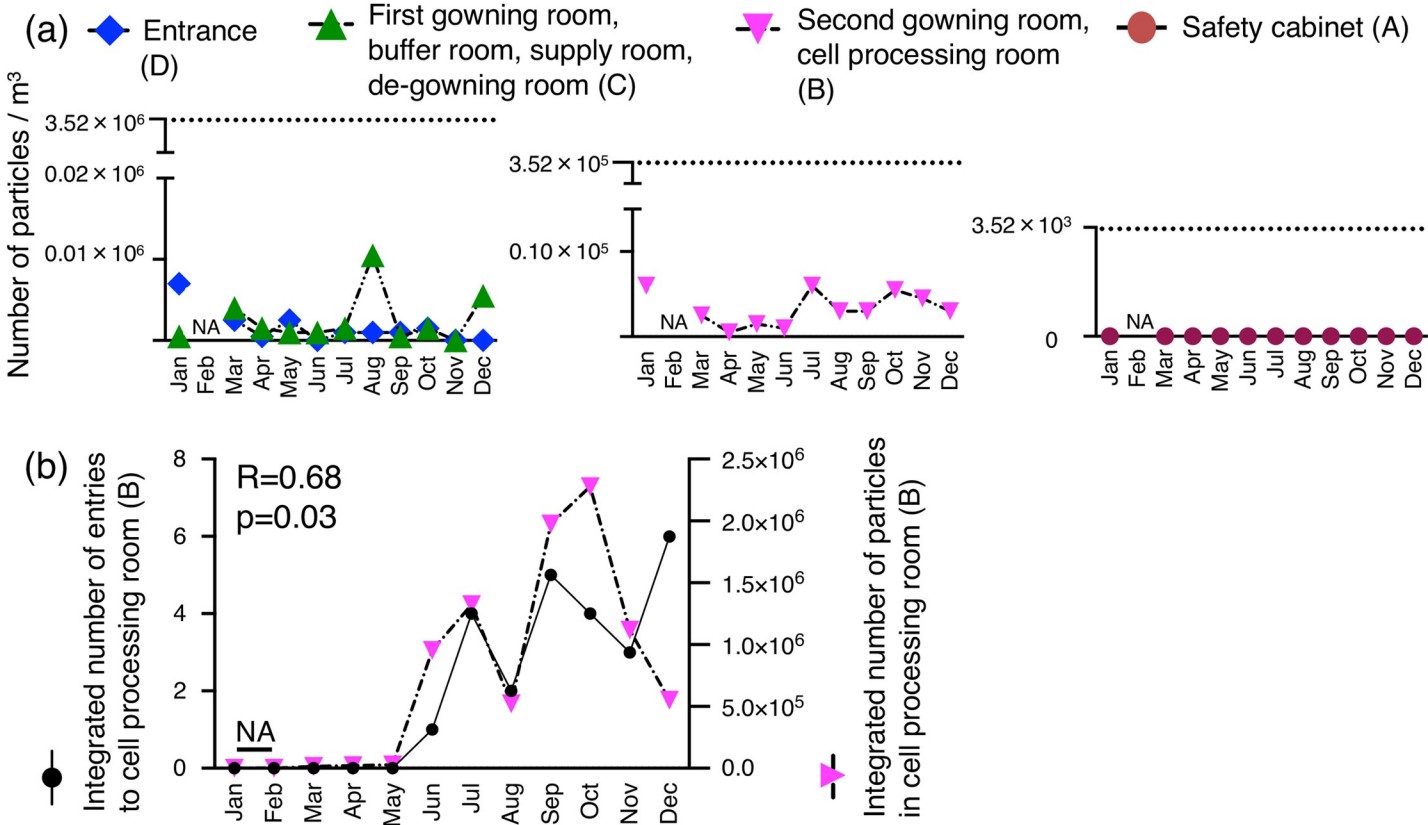

**Fig 3. Monthly transitions in the number of particles per room and the relationship between this and the number of entries to the cell-processing room.** (a) Monthly changes in the number of particles ($\geq$ 0.5μm) in the room classified by each cleanliness level. The data in cleanliness-controlled areas were shown as the maximum value in each room. The dotted line indicates the acceptable limit for each room classified at each cleanliness level. (b) Relationship between integrated number of entries to the cell-processing room (Grade B) and integrated number of particles in the cell-processing room (Grade B). p = 0.03 by Pearson's correlation coefficient (R = 0.68). NA: not applicable.

it was not measured (Fig 4B). The number of colonies was well below the acceptable limit, which was under <1 for Grade A, 10 for Grade B and 100 for Grade C and D. The entrance times into Grade B did not correlate with the number of airborne microorganisms (S2 Fig). PCR products from the colonies were sequenced directly, and seven bacteria, including four gram-negative and two gram-positive bacteria, along with two fungi, were identified (S2 Table). The gross appearance of the colonies varied in color, morphology, and size (Fig 4C). Positive colony detection rates were significantly higher at 55% or higher humidity with an odds ratio of 2.47 (Fig 4D).

## Monitoring of harmful insects

The total number of harmful insects per trap in one year shows that the majority were trapped in the corridor outside the CPF (Fig 5A). The total number of harmful insects per trap peaked in the corridor outside the CPF in April and August and in the laboratory outside the CPF in August (Fig 5B). The total number of insects per trap in the corridor outside the CPF was significantly increased when temperature outside the building was higher than 15˚C (S3 Fig). Even in controlled areas, a small number of insects was constantly trapped between April and December (Fig 5B). Of the total number of insects trapped, the representative six species accounted for 92% (Fig 5C and 5D). The species of trapped insects varied by area, and the

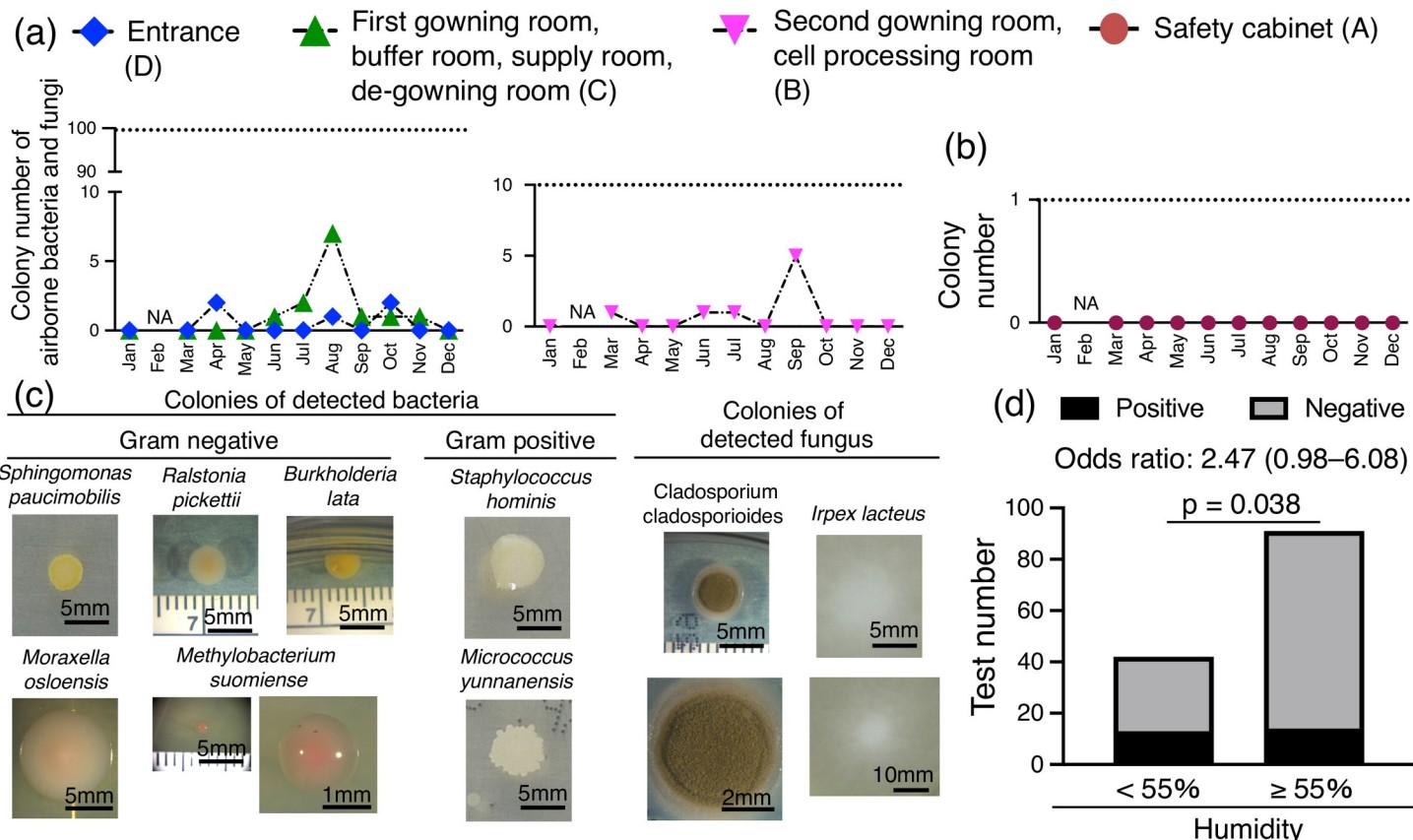

**Fig 4. Monitoring of microorganisms and identification of species.** (a) Monthly transition of colony number of airborne bacteria and fungi. Settle plate and glove print tests were also performed by using a soybean-casein digest agar in the Grade A area. The data in cleanliness-controlled areas were shown as the maximum value in each room. The dotted line indicates the allowable reference value in each room classified by each cleanliness level. NA: not applicable. (b) Monthly transition in falling bacteria and bacteria attached to gloves. (c) Representative colonies of detected airborne bacteria and fungus in soybean-casein digest agar. (d) Relationship between humidity and colony detection rate. The number of tests and the number of tests in which positive colonies were detected are shown for humidity of 55% or more and humidity less than 55%. p = 0.038 by the chi-square test.

peaks of their numbers varied by the seasons (Fig 5E and 5F). *Psocoptera* accounted for 41.1% and peaked in August, and only Socoptera was trapped in Grade B. *Acari* accounted for 40.9% and peaked in April, and they were trapped during summer in all the areas except for Grade B. *Araneae*, *Collembola*, *Psychodidae*, and *Blattidae* accounted for 2–3% each. The others consisted of various species, such as *Hermann* (Fig 5D).

## Cell production and sterility

In the one year we assessed, six cell products for transplantation were cultured in this CPF. Their sterilities were guaranteed by the sterility test (S3 Table).

## Discussion

Although cell products cannot be sterilized, their sterility must be ensured for patient safety. To achieve the sterility of cell products, elimination of the risk of extrinsic contamination during the process is necessary. The assessment of environmental risk factors inside and outside CPFs will be effective to eliminate the contamination. However, these risk factors fluctuate depending on the season, CPF location, and human intervention. In the present study, we

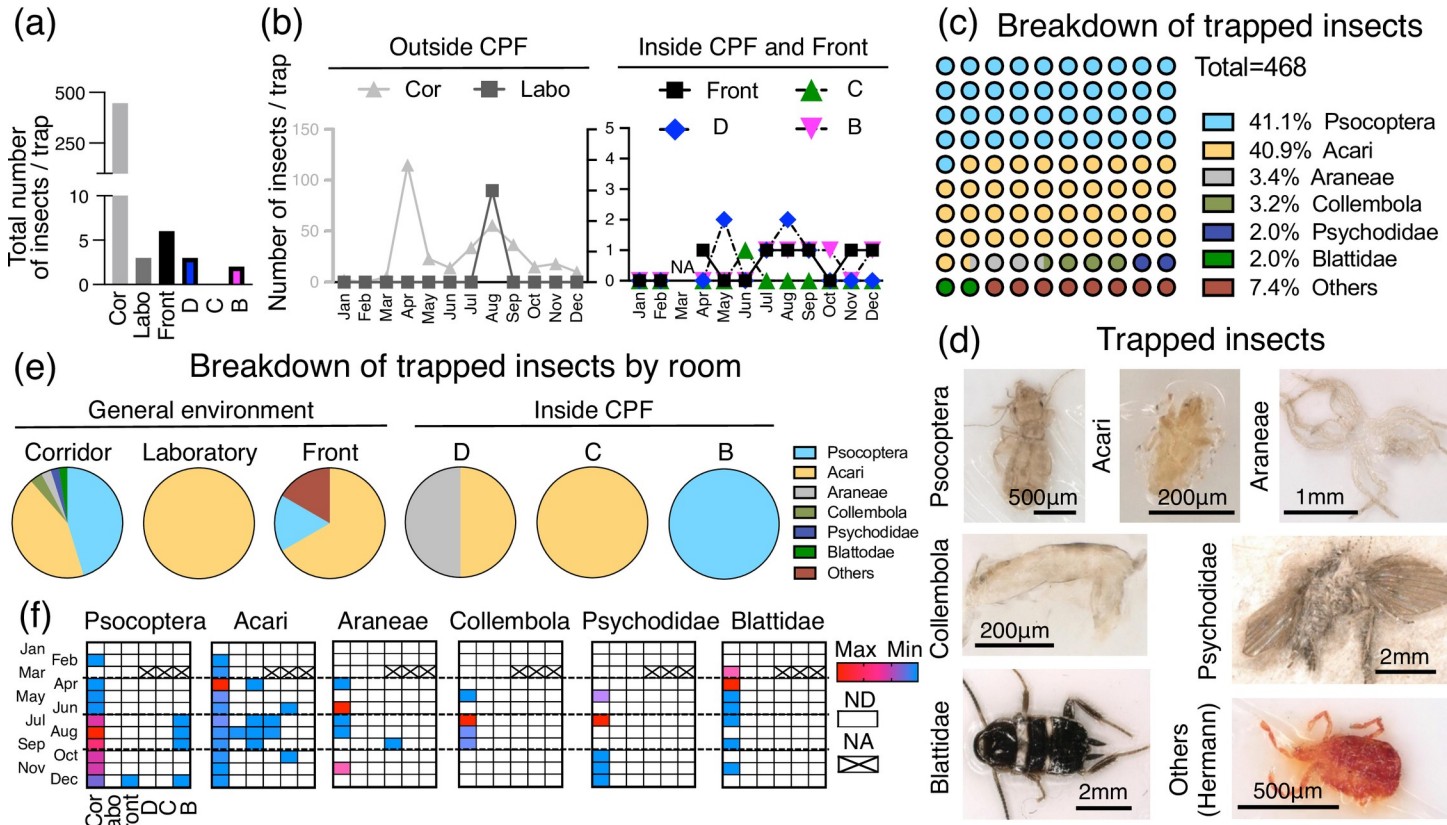

**Fig 5. Monitoring of harmful insects.** (a) Total number of insects in one year per trap. (b) Monthly transition of insects trapped outside or inside the CPF. The number of insects is shown as maxima per trap for each month. Left y-axis indicates the number of insects per trap in the corridor outside the CPF. Right y-axis indicates the number of insects per trap in the laboratory outside the CPF. (c) Breakdown of trapped insects. One dot represents 1%. (d) Representative macroscopic images of the trapped insects. (e) Breakdown of trapped insects by room. (f) Heat map for monthly transitions of each representative insect per trap in each room. GE: general environment. Cor: corridor and workspace outside the CPF. Labo: laboratory outside the CPF. Front: in front of the cell-processing facility outside the CPF. Grade D: entrance. Grade C: first gowning room, buffer room, supply room, and de-gowning room. Grade B: second gowning room and cell-processing room. ND: not detected. NA: not applicable.

assessed environmental factors in a CPF and its adjacent GE throughout the year. Their effects on the sterility of cell products were also assessed. Our data suggested that various environmental factors could be a risk to the safety of cell products, and some of the factors interacted with each other, such as the relationship between humidity and colony number of airborne bacteria/fungi.

The cleanliness-controlled area in our CPF was maintained at a constant temperature, but the humidity fluctuated by the season. The equipment for humidity control is expensive and usually not set up in academic institutions. However, high humidity is usually the most influential factor for the growth of bacteria, fungi, and harmful insects [14–16]. In the present study, positive colony detection rates for bacteria and fungus from airborne microorganisms were significantly higher at humidity of 55% or higher. The fungi identified in this study, such as *Irpex lacteus*, are known to grow faster under humid conditions [14], and their proliferation could affect other factors, such as the number of *Psocoptera* that feed on fungi [15, 17]. Therefore, high humidity could be a major threat to the safety of cell products, and the control of humidity is an important issue for CPF management. It may thus be a solution to introduce a simple, temporal humidity control device in humid seasons.

In this study, the number of particles ($\geq$ 0.5μm) was within the acceptable range in all cleanliness-controlled areas throughout the year. Depending on the number of entrances made, the integrated number of particles in Grade B increased; however, the number of entries to the cell-processing room (Grade B) did not directly increase the number of airborne microorganisms. The particles in clean rooms have been reported to be dust from clothes or human skin flakes [18]. In addition, most microorganisms attach to dusts, skin flakes, and water droplets [18]. Ten to 1,000 suspended aerosol particles are considered to contain one microorganism [19, 20]. One report demonstrated that the number of particles correlated with that of airborne microorganisms in an operating theater and an ICU [21, 22]. On the other hand, some studies have reported that the number of particles do not correlate with the number of microorganisms [23–25]. Our results support the latter finding. Nevertheless, during the processing of clinical cell products, it will be essential to minimize the number of entries.

Environmental microorganisms were found in the humid season in Grade D, Grade C, and Grade B but were not detected in Grade A. The detection of environmental microorganisms is important to risk management in CPFs. For example, the generation of airborne microorganisms in controlled areas could occur due to structural problems in CPFs associated with outdoor inflow and human problems caused by carrying in materials. Therefore, the need for the identification of microorganisms is described in pharmacopoeias such as Japanese Pharmacopoeia, European Pharmacopoeia, and United States Pharmacopoeia. The microorganisms isolated in this study were reported to live in the GE [26–28]. However, there were multiple candidate species from BLAST, and they had still not been identified as one species. Our results showed that the gross appearance of the colonies varied in color, morphology, and size, which might be useful for the estimation of species to decide countermeasures against them. When the same colonies are detected, it will be easy to determine the cause and appropriate countermeasure. On the other hand, when the colonies are newly detected, the proper countermeasures, such as identification of species, consideration of possible causes, and selection of the optimal disinfection methods, are needed. The characteristics of each species (bacterium or fungi, resistance to disinfectants, high or low growth rate, etc.) is useful for CPF management policy.

We monitored microorganisms and identified seven bacteria, including four gram-negative bacteria and two gram-positive bacteria. *Sphingomonas paucimobilis*, which lives widely in the environment, is a highly adaptable bacteria that was once recognized in a spacecraft meal [29, 30]. *Ralstonia pickettii* [31] and *Burkholderia lata* [32] have been reported to live in chlorhexidine disinfectant and are harmful environmental bacteria. *Moraxella osloensis*[33] is known as an upper respiratory tract–resident bacteria. This bacterium is a major environmental microorganisms, producing 4-methyl-3-hexenoic acid (4M3H), which is a cause of malodor. *Methylobacterium suomiense* was recognized in small colonies in our study. A single detection of *Methylobacterium suomiense* is unlikely to be an environmental risk because of their low growth rates. However, they are resistant to chlorine in tap water and surfactants, resulting in the continuous and widespread detection of them even after treatment with usual disinfection alone. To disinfect them, hot water at 65˚C or higher is necessary [34]. *Staphylococcus hominis* [35], one of the gram-positive bacteria, is also reported to be important as a nosocomial pathogen. Micrococcus yunnanensis [36] was isolated as environmental bacteria.

We also identified two fungi in the CPF: *Cladosporium cladosporioides*, which has been reported as a cause of sick building syndrome [37], and *Cladosporium cladosporioides*, which attaches to air conditioners [38] and present in the environment for a long time. If isolated multiple times in CPFs, the decontamination of the entire CPF will be required. *Irpex lacteus* [39] is a white rot fungus distributed in the environment, such as in dead broadleaf trees.

Knowing these characteristics of detected microorganisms makes it possible to infer their entry routes into CPFs and to plan disinfection strategies.

Isolated species in the present study were known to have various origins and to inhabit, not only the environment, but also human skin [10, 40]. In fact, microorganisms derived from human skin were detected in stem cell banks in Spain [41]. In addition, Sandle et al. reported 6,729 isolated colonies of the bacteria in the Grade A and B clean areas [42]. In their review paper, the colonies of *Micrococcus* and *Staphylococcus*, skin resident bacteria, were detected most commonly, accounting for 50% of total colonies. The second-most common genera were species of *Bacillus*, an environmental bacterium, accounting for 13%. However, other reports have demonstrated types of detected microorganisms to be rich in variety [43–45]. The different microbiome may result from the difference in climate and processing products. In a humid environment, such as in Singapore, various microorganisms tend to be detected [46]. Resistant bacteria have also been found in antibiotic-producing facilities [11, 47]. To maintain CPFs properly, facility-specific evaluations based on local climatic characteristic and the processing products are very important.

In the corridor outside the CPF, many harmful insects were trapped in spring, summer, and autumn. Though on a smaller scale, similar results were found within the CPF. Such seasonal variability is useful for predicting insect number within the CPF. Insect intrusions into clean areas are known to be caused by humans or by decrepit facilities [48]. Since our CPF has been completed for fewer than five years, it is highly probable that insect invasion was accompanied by the movement of humans and materials. *Psocoptera*, commonly known as booklice [49], adhere to cardboard and clothes. *Psocoptera* were not trapped in Grade D or Grade C but in Grade B; it is thus highly likely that they were brought in with construction materials. Insect intrusion into a CPF may contaminate cell products because many harmful microorganisms attach to them [50–52]. Therefore, it will be important to periodically assess the number and characteristics of intruding insects, which can be affected by the number of entrance times, the age of the CPF, and appropriate feedback on risk management policy.

Though bacteria, fungi, and insects were detected in Grade B–D, no contamination was detected in the cell products during this study. One reason may be that bacteria, fungi, or insects were not detected in Grade A. Even if contamination occurs in the cell products, microorganisms can be detected by a sterility test of the cell products. Although this study focused on environmental factors, management of the entire process, including the detection of contamination and assurance of quality, is also important to realize the safety of cell products.

We faced three limitations in this study. First, larger particles ($\geq$ 10 or 20 μm) were not evaluated because our CPF was completed only five years ago and we thought that there was no deterioration of the building. Measurement of large particles is considered useful for the detection of building deterioration [21, 22]. Second, the difference of the distribution of microorganisms by season and room was not fully analyzed. If microorganisms are brought in from the outside by clothes, improving the buffer room can be a countermeasure. If they are derived from human skin, changing clean room clothing or modifying entrance procedures may be useful. For them, more detailed analysis of the distribution of microorganism species in multiple rooms will be needed. Third, the classification of room cleanliness did not follow ISO standards or the Japanese Pharmacopoeia. In Japan, regenerative medicine using cell products is regulated by two laws: the Pharmaceuticals, Medical Devices and Other Therapeutic Products Act and the Act on the Safety of Regenerative Medicine (Safety Act), which are applied in the field of medical care and clinical research, respectively [6, 7]. Our clinical study was performed using Safety Act standards, not ISO or Pharmacopoeia standards.

In conclusion, this study assessed many environmental risks inside and outside a CPF throughout the year. Environmental factors varied with the seasons. The temperature in the CPF was constant, but the humidity fluctuated greatly depending on the season. The number of particles correlated with the number of entrance times. Various microorganisms were identified in the CPF, and high humidity increased their occurrence ratio. The types of trapped insects varied by CPF area. More insects were trapped in the season when outside temperature was higher than 15˚C. Although many risks were found in the environment around the CPF, the sterility of the aseptic operation area and processed cells was maintained. These results provide useful data for taking the appropriate steps to keep entire CPFs clean.

## Supporting information

**S1 Fig. Relationship between integrated number of entries to the cell-processing room (Grade B) and integrated number of particles ($\geq$ 5µm) in the cell-processing room (Grade B).** p = 0.048 by Pearson's correlation coefficient (R = 0.64). NA: not assigned.
(PDF)

**S2 Fig. The number of airborne microorganisms and entrance times to the cell-processing room (Grade B).**
(PDF)

**S3 Fig. The effect of temperature outside the building on the total number of insects per trap in the corridor outside the CPF.** Data are shown by median with interquartile range. p values were calculated by the Mann Whitney U-test.
(PDF)

**S1 Table. Environment definition.**
(PDF)

**S2 Table. Sequence data of identified bacterium.**
(PDF)

**S3 Table. Sterility in cell processing products.**
(PDF)

## Acknowledgments

We would like to thank Ms. Jun Kusano and Ms. Akiko Hoshikawa for the management of the Center for Cell Therapy, Ms. Mika Watanabe and Ms. Kimiko Takanashi for the management of our laboratory, and Joe Lewis for English editing.

## Author Contributions

**Conceptualization:** Mitsuru Mizuno.

**Investigation:** Mitsuru Mizuno, Ayako Tsuji, Naomi Kojima, Ken Watanabe, Norio Shimizu.

**Project administration:** Mitsuru Mizuno.

**Supervision:** Hisako Katano, Tomohiro Morio, Ichiro Sekiya.

**Validation:** Ichiro Sekiya.

**Visualization:** Mitsuru Mizuno.

**Writing – original draft:** Mitsuru Mizuno.

**Writing – review & editing:** Mitsuru Mizuno, Kentaro Endo, Ichiro Sekiya.

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
