## [Decision Letter · Decision Letter 0]

23 Jun 2020

PONE-D-20-15060

The environmental risk assessment of cell-processing facilities for cell therapy in a Japanese academic institution

PLOS ONE

Dear Dr. Sekiya,

Thank you for submitting your manuscript to PLOS ONE. After careful consideration, we feel that it has merit but does not fully meet PLOS ONE’s publication criteria as it currently stands. Therefore, we invite you to submit a revised version of the manuscript that addresses the points raised during the review process.

We look forward to receiving your revised manuscript.

Kind regards,

Raghvendra Bohara

Academic Editor

PLOS ONE

Additional Editor Comments:

 Minor revision- check for typos throughout.

3. Your ethics statement must appear in the Methods section of your manuscript. If your ethics statement is written in any section besides the Methods, please move it to the Methods section and delete it from any other section. Please also ensure that your ethics statement is included in your manuscript, as the ethics section of your online submission will not be published alongside your manuscript.

Reviewers' comments:

Reviewer's Responses to Questions

**Comments to the Author**

1. Is the manuscript technically sound, and do the data support the conclusions?

Reviewer #1: Yes

Reviewer #2: Yes

2. Has the statistical analysis been performed appropriately and rigorously? 

Reviewer #1: Yes

Reviewer #2: Yes

3. Have the authors made all data underlying the findings in their manuscript fully available?

Reviewer #1: Yes

Reviewer #2: Yes

4. Is the manuscript presented in an intelligible fashion and written in standard English?

Reviewer #1: Yes

Reviewer #2: Yes

5. Review Comments to the Author

Reviewer #1: This manuscript is well written, with a clear and concise study design. The findings provide a detailed insight into the environmental risk factors associated with sterility of cell-processing facilities. The methodology for this study is clearly explained. The recommendations to mitigate environmental risks are made clear in the discussion of this study. The authors clarify the importance and impact of this study to influence CPF management policy to ensure cell-product quality. It is a reassuring study to highlight that while all products were produced aseptically and maintained sterility, further precautions are being taken to maintain the high quality procedures in CPFs.

Page 7, line 5 - “below 14 and above 15” – rephrase to include 14-15°C

Page 7, line 16 - “Absence of pathogens” – “rephrase to presence or absence”

“The number of particles was well below the acceptable limit for each room classified at each cleanliness level” – state, unclear

“The number of colonies was well below the acceptable limit for each room classified at each cleanliness level” – while these values are given in the figures and outlined in supplementary, it would be helpful to include in the text also for reference.

Figures are correctly labelled and clearly displayed.

Figure 5. The breakdown of trapped insects is unclear (e + f). Inside CPF in (e) states Psocoptera was found in B but (f) fails to highlight this finding. Same is true for D,C and B throughout.

Reviewer #2: The authors of the current research manuscript have summarized the environmental risk assessment of cell-processing facilities (CPF) for cell therapy in Japanese academic institution. They have mainly presented the methods for monitoring the temperature, humidity, particle number, colony number of microorganisms, bacteria, fungi, and harmful insects in and around CPF monthly over one year and is well written. Figures are clear and properly labelled. Figure legends are appropriate and well written. Overall, the manuscript is well written and major areas on the appropriate steps to keep entire CPFs clean are covered. The manuscript is authoritative. This study has clearly demonstrated the importance of the safety of cell products and environmental risk factors with ongoing clinical study and it is reassuring. However, there are some minor improvements are required before accepting the manuscript for publication.

Minor Comments:

1. Keep the ‘P’ letter used for statistical analysis constant, e.g. Method section: Statistical analysis and supplementary figure 3

2. Page 20, Line 3: The size of integrated number of particles is mentioned ‘> 0.5 µm’ however rest everywhere it is ‘≥ 0.5 µm’. Please rectify this.

3. Please provide reference here: Page 3, Introduction section, 14th-16th line- “The risk of extrinsic contamination can be affected by the location of the CPF, the environment inside or outside the CPF, and the degree of human intervention during the process.”

6. PLOS authors have the option to publish the peer review history of their article (what does this mean?). If published, this will include your full peer review and any attached files.

Reviewer #1: No

Reviewer #2: No

---

## [Author Response · Author response to Decision Letter 0]

5 Jul 2020

Manuscript No. PONE-D-20-15060

July 6, 2020

PLOS ONE

Academic Editor

Dear Dr. Raghvendra Bohara,

 Thank you for your comments on June 24, 2019. The title of our manuscript is “The environmental risk assessment of cell-processing facilities for cell therapy in a Japanese academic institution.” Based on the valuable comments provided by the two reviewers, we have rewritten the manuscript, revised three figures, and added one supplemental figure. We believe these revisions address the reviewers’ comments and that the paper now meets the journal’s publication standards. We would very much appreciate having the paper reconsidered for publication in PLOS ONE.

Additional Editor Comments:

and

Author’s response:

 We thank the editor for the detailed proofreading of this study. In the revised manuscript, we have addressed all the concerns raised regarding the original manuscript. We have ensured that all heading levels are clearly indicated in the manuscript text, and we have modified the names of the supplementary figures to S1 Figure, S2 Figure, etc.

Author’s response:

 We removed the sentence from the manuscript as follows:

 Since our CPF has been completed for fewer than five years, it is highly probable that insect invasion was accompanied by the movement of humans and materials. Psocoptera, commonly known as booklice [49], adhere to cardboard and clothes. Psocoptera were not trapped in Grade D or Grade C but in Grade B; it is thus highly likely that they were brought in with construction materials.

3. Your ethics statement must appear in the Methods section of your manuscript. If your ethics statement is written in any section besides the Methods, please move it to the Methods section and delete it from any other section. Please also ensure that your ethics statement is included in your manuscript, as the ethics section of your online submission will not be published alongside your manuscript.

Author’s response:

 We added the following section to the manuscript:

Ethics approval and consent to participate

 This clinical study (RM2018-006) was approved by the certified special committee for regenerative medicine at the Tokyo Medical and Dental University (committee reference number: NA8140003) and submitted a plan to the Minister of Health, Labor, and Welfare (MHLW, PB3160032). This clinical study complied with the Helsinki Declaration.

Reviewer #1:

 This manuscript is well written, with a clear and concise study design. The findings provide a detailed insight into the environmental risk factors associated with sterility of cell-processing facilities. The methodology for this study is clearly explained. The recommendations to mitigate environmental risks are made clear in the discussion of this study. The authors clarify the importance and impact of this study to influence CPF management policy to ensure cell-product quality. It is a reassuring study to highlight that while all products were produced aseptically and maintained sterility, further precautions are being taken to maintain the high quality procedures in CPFs.

Author’s response:

 We thank Reviewer #1 for the thoughtful comments on this study. In this revised manuscript, we have addressed all the concerns raised regarding the original manuscript.

Page 7, line 5 - “below 14 and above 15” – rephrase to include 14-15°C

Author’s response: We rewrote the manuscript and S3 Figure as follows:

 The number of insects per trap in the corridor outside the CPF was compared below 15 °C and 15 °C or more outside the building by the Mann Whitney U-test.

S3 Figure.

To address Reviewer #1’s comment, we noticed the mistake in Figure 4d and modified it as follows:

Page 7, line 16 - “Absence of pathogens” – “rephrase to presence or absence”

Author’s response:

 The original sentence was “After 14 days, the absence of pathogens was visually evaluated.” This sterility test cannot deny the existence of another pathogen-like virus, so we corrected the sentence as follows:

 “After 14 days, the presence or absence of microorganisms was visually evaluated.”

“The number of particles was well below the acceptable limit for each room classified at each cleanliness level” – state, unclear

“The number of colonies was well below the acceptable limit for each room classified at each cleanliness level” – while these values are given in the figures and outlined in supplementary, it would be helpful to include in the text also for reference.

Figures are correctly labelled and clearly displayed.

Author’s response:

 To address this comment, we modified the sentences in the Results section as follows:

 “The number of particles was well below the acceptable limit, which was under 3,520 for grades A and B, 35,200 for grade C, and 3,520,000 for grade D.”

 “The number of colonies was well below the acceptable limit, which was under 1 for grade A, 10 for Grade B, and 100 for grades C and D.”

Figure 5. The breakdown of trapped insects is unclear (e + f). Inside CPF in (e) states Psocoptera was found in B but (f) fails to highlight this finding. Same is true for D,C and B throughout.

Author’s response:

We modified Figure 5f to improve the visibility of trapped insects as follows:

Figure 5f

Reviewer #2: The authors of the current research manuscript have summarized the environmental risk assessment of cell-processing facilities (CPF) for cell therapy in Japanese academic institution. They have mainly presented the methods for monitoring the temperature, humidity, particle number, colony number of microorganisms, bacteria, fungi, and harmful insects in and around CPF monthly over one year and is well written. Figures are clear and properly labelled. Figure legends are appropriate and well written. Overall, the manuscript is well written and major areas on the appropriate steps to keep entire CPFs clean are covered. The manuscript is authoritative. This study has clearly demonstrated the importance of the safety of cell products and environmental risk factors with ongoing clinical study and it is reassuring. However, there are some minor improvements are required before accepting the manuscript for publication.

Authors’ response:

 We appreciate the meaningful, favorable comments by Reviewer #2 on this study. In this revised manuscript, we have addressed all the concerns raised regarding the original manuscript.

Reviewer #2 Minor Comments:

1. Keep the ‘P’ letter used for statistical analysis constant, e.g. Method section: Statistical analysis and supplementary figure 3

Author’s response:

We rewrote the manuscript, figures, and figure legends as follows:

Figure 4d

S3 Figure

Legend for Figure 3:

 Monthly transitions in the number of particles per room and the relationship between this and the number of entries to the cell-processing room. (a) Monthly changes in the number of particles (> 0.5 μm) in the room classified by cleanliness level. The data in cleanliness-controlled areas were shown as the maximum value in each room. The dotted line indicates the acceptable limit for each room classified at each cleanliness level. (b) Relationship between integrated number of entries to the cell-processing room (Grade B) and integrated number of particles in the cell-processing room (Grade B). p = 0.03 by Pearson’s correlation coefficient (R = 0.68). NA: not applicable.

Legend for Figure 4:

 Monitoring of microorganisms and identification of species. (a) Monthly transition of colony number of airborne bacteria and fungi. Settle plate and glove print tests were also performed using a soybean-casein digest agar in the Grade A area. The data in cleanliness-controlled areas were shown as the maximum value in each room. The dotted line indicates the allowable reference value in each room classified by cleanliness level. NA: not applicable. (b) Monthly transition in falling bacteria and bacteria attached to gloves. (c) Representative colonies of detected airborne bacteria and fungus in soybean-casein digest agar. (d) Relationship between humidity and colony detection rate. The number of tests and the number of tests in which positive colonies were detected are shown for humidity of 55% or more and less than 55%. p = 0.038 by the chi-square test.

Legend for S1 Figure:

 Relationship between integrated number of entries to the cell-processing room (Grade B) and integrated number of particles (≥ 5μm) in the cell-processing room (Grade B). p = 0.048 by Pearson’s correlation coefficient (R = 0.64). NA: not assigned.

Legend for S3 Figure:

 The effect of temperature outside the building on the total number of insects per trap in the corridor outside the CPF. Data are shown by median with interquartile range. p values were calculated by the Mann Whitney U-test.

2. Page 20, Line 3: The size of integrated number of particles is mentioned ‘> 0.5 µm’ however rest everywhere it is ‘≥ 0.5 µm’. Please rectify this.

Author’s response:

We rewrote the legend for Figure 3 as follows:

Figure 3. Monthly transitions in the number of particles per room and the relationship between this and the number of entries to the cell-processing room. (a) Monthly changes in the number of particles (≥ 0.5 μm) in the room classified by cleanliness level. The data in cleanliness-controlled areas were shown as the maximum value in each room. The dotted line indicates the acceptable limit for each room classified at each cleanliness level. (b) Relationship between integrated number of entries to the cell-processing room (Grade B) and integrated number of particles in the cell-processing room (Grade B). p = 0.03 by Pearson’s correlation coefficient (R = 0.68). NA: not applicable.

3. Please provide reference here: Page 3, Introduction section, 14th-16th line- “The risk of extrinsic contamination can be affected by the location of the CPF, the environment inside or outside the CPF, and the degree of human intervention during the process.”

Author’s response:

We added two references and rewrote the lines in question as follows:

“The risk of extrinsic contamination can be affected by the location of the CPF, the environment inside or outside the CPF, and the degree of human intervention during the process [10, 11].”

[10.] Prussin AJ, Marr LC. Sources of airborne microorganisms in the built environment. Microbiome. 2015;3(1):78. Epub 2015/12/24. doi: 10.1186/s40168-015-0144-z. PubMed PMID: 26694197; PubMed Central PMCID: PMCPMC4688924.

This review article explains that sources of microbial bioaerosols in the built environment may include humans; pets; plants; plumbing systems; heating, ventilation, and air-conditioning systems; mold; resuspension of settled dust; and outdoor air.

[11.] Hamdy AM, El-Massry M, Kashef MT, Amin MA, Aziz RK. Toward the drug factory microbiome: Microbial community variations in antibiotic-producing clean rooms. OMICS: A Journal of Integrative Biology. 2018;22(2):133–44. Epub 2017/09/06. doi: 10.1089/omi.2017.0091. PubMed PMID: 28873001.

This research article reports that the microbial composition of a nonantibiotic drug product was highly affected by the use of water, environmental conditions during the production process, the presence of personnel, and the type of product.

Sincerely yours,

Ichiro Sekiya, MD, PhD

Director, Center for Stem Cell and Regenerative Medicine

Professor, Department of Applied Regenerative Medicine

Tokyo Medical and Dental University (TMDU)

1-5-45 Yushima, Bunkyo-ku, Tokyo, 113-8510, Japan

Tel: +81-3-5803-4017, Fax: +81-3-5803-0192, E-mail: sekiya.arm@tmd.ac.jp

---

## [Decision Letter · Decision Letter 1]

10 Jul 2020

The environmental risk assessment of cell-processing facilities for cell therapy in a Japanese academic institution

PONE-D-20-15060R1

Dear Dr. Sekiya,

We’re pleased to inform you that your manuscript has been judged scientifically suitable for publication and will be formally accepted for publication once it meets all outstanding technical requirements.

Kind regards,

Raghvendra Bohara

Academic Editor

PLOS ONE

Reviewers' comments:

Reviewer's Responses to Questions

**Comments to the Author**

1. If the authors have adequately addressed your comments raised in a previous round of review and you feel that this manuscript is now acceptable for publication, you may indicate that here to bypass the “Comments to the Author” section, enter your conflict of interest statement in the “Confidential to Editor” section, and submit your "Accept" recommendation.

Reviewer #1: All comments have been addressed

Reviewer #2: All comments have been addressed

2. Is the manuscript technically sound, and do the data support the conclusions?

Reviewer #1: Yes

Reviewer #2: Yes

3. Has the statistical analysis been performed appropriately and rigorously? 

Reviewer #1: Yes

Reviewer #2: Yes

4. Have the authors made all data underlying the findings in their manuscript fully available?

Reviewer #1: Yes

Reviewer #2: Yes

5. Is the manuscript presented in an intelligible fashion and written in standard English?

Reviewer #1: Yes

Reviewer #2: Yes

6. Review Comments to the Author

Reviewer #1: All comments regarding the original submission have been addressed and the authors now present a robust manuscript.

Reviewer #2: Accept

7. PLOS authors have the option to publish the peer review history of their article (what does this mean?). If published, this will include your full peer review and any attached files.

Reviewer #1: No

Reviewer #2: No

---

## [Editor Report · Acceptance letter]

16 Jul 2020

PONE-D-20-15060R1 

The environmental risk assessment of cell-processing facilities for cell therapy in a Japanese academic institution 

Dear Dr. Sekiya:

I'm pleased to inform you that your manuscript has been deemed suitable for publication in PLOS ONE. Congratulations! Your manuscript is now with our production department. 

Kind regards, 

on behalf of

Dr. Raghvendra Bohara 

Academic Editor

PLOS ONE